# Hidden diversity of Trypanosomatidae (Protozoa: Kinetoplastea) in bats from an urban park in Brazil

**Mariana Alves Lima**[1], **Marcela Elisa Vaz**[1], **Jennifer Emanuelle Ferreira**[2], **Ana Cristina Vianna Mariano da Rocha Lima**[1], **Débora Cristina Capucci**[1], **Sônia Aparecida Talamoni**[2], **Felipe Dutra-Rêgo**[1☯*], **José Dilermando Andrade Filho**[1☯*]

1 Leishmaniasis Group, René Rachou Institute, Oswaldo Cruz Foundation, Belo Horizonte, Minas Gerais, Brazil, 2 Graduate Program in Biodiversity and Environment, Pontifical Catholic University of Minas Gerais, Belo Horizonte, Minas Gerais, Brazil

☯ These authors contributed equally to this work.
* jose.andrade@fiocruz.br (JDAF); felipedutra04@hotmail.com (FD-R)

## Abstract

Trypanosomatids are obligate flagellated parasites, with the genera *Leishmania* and *Trypanosoma* acting as etiological agents of significant diseases such as leishmaniasis and Chagas disease. Although ecological studies have increasingly highlighted the role of bats as potential reservoirs of these parasites, the diversity of trypanosomatids in urban bat populations remains poorly understood. This study investigates the occurrence and diversity of Trypanosomatidae in bats from Mangabeiras Municipal Park (MMP), an urban park in Belo Horizonte, Minas Gerais, Brazil, a region of ecological interest due to the prior detection of *Leishmania* in sand flies. A total of 56 bats representing seven species were captured, and 149 biological samples (blood, tissues, and organs) were analyzed using NNN/LIT culture medium. Contamination was reported in 32.2% of the samples, while 67.8% yielded negative results with no growth of trypanosomatids. Detection of trypanosomatids was achieved using the V7V8 Nested-PCR technique, revealing positive results in nine bats: *Artibeus lituratus* (*Leishmania infantum*, *Trypanosoma* sp. Neobat 3), *Anoura caudifer* (*Trypanosoma* sp. Neobat 4), *Carollia perspicillata* and *Glossophaga soricina* (*Leishmania infantum*), *Sturnira lilium* (*Trypanosoma* sp. Neobat 3), and *Platyrrhinus lineatus* (mixed infection with *Leishmania infantum* and *Leishmania braziliensis*). The integrity of the extracted DNA was confirmed through the amplification of *cytb* and *gamma-actin* genes. By expanding knowledge of trypanosomatid diversity in urban bats, this study highlights the ecological and epidemiological relevance of bats as hosts and underscores the need for targeted surveillance to assess their role in pathogen transmission dynamics.

## Introduction

The order Trypanosomatida comprises obligate flagellated endoparasites, with the genera *Leishmania* and *Trypanosoma* being particularly noteworthy. *Leishmania* is responsible for visceral (VL) and cutaneous leishmaniasis (CL), while *Trypanosoma* includes parasites associated with Chagas disease in the Americas (*Trypanosoma cruzi*) and sleeping sickness in Africa

**Data availability statement:** All relevant data are within the manuscript.

**Funding:** JDAF, Grant number 314260/2023-4, Conselho Nacional de Desenvolvimento Científico e Tecnológico, https://www.gov.br/cnpq/pt-br, The sponsor have no role in the study design, data collection and analysis, decision to publish, or preparation of the manuscript.

**Competing interests:** The authors have declared that no competing interests exist.

(*Trypanosoma brucei*) [1]. Trypanosomatids are transmitted to vertebrate hosts through various mechanisms. *Leishmania* is transmitted via the bite of sand flies during blood feeding [2], while *T. cruzi* is primarily transmitted through the release of triatomine feces and urine (Hemiptera: Reduviidae: Triatominae) during hematophagy, although other transmission routes have also been reported [3]. Trypanosomatids are among the most extensively studied protists and, based on their life cycle, can be subdivided into two non-taxonomic groups: monoxenous (involving a single host) and heteroxenous (involving two or more hosts, one of which is a vector). The phylogenomic structure of these parasites, typically analyzed through multiple protein-coding genes, has revealed a complex and heterogeneous relationship among the genera [4].

In general, trypanosomatids exhibit ecological associations with a wide range of hosts, including humans [5]. However, enzootic cycles associated with these pathogens are primarily linked to mammals in wild areas [6,7]. The high susceptibility of bats to infection by various trypanosomatids has been extensively studied [8,9], and in some cases, bats have been suggested as potential reservoirs, particularly for protozoa of the genus *Trypanosoma* and possibly *Leishmania* [8,10,11]. Given the diversity and potential for mixed infections in bat populations, the use of sensitive and specific detection methods, such as PCR, is essential for analyzing biological samples that may present low and heterogeneous parasite loads [12–14]. The significance of these animals as natural hosts for various pathogens is further amplified by their ecological habits, which span vast geographical regions and their adaptability to diverse environments, including urban areas [15], potentially facilitating transmission, particularly to humans.

Despite growing evidence of trypanosomatid infections in bats, most studies have focused on rural or sylvatic settings [14,16,17], with limited investigations into urban environments where bats frequently interact with humans and domestic animals. Recent studies have increasingly underscored the role of urban areas in harboring diverse trypanosomatid species, both in endemic and non-endemic regions, providing critical insights for guiding health authorities in epidemiological surveillance efforts [13,18]. Urban parks, in particular, may serve as hotspots for pathogen transmission due to the overlap of human, domestic, and wildlife populations. This study investigates the occurrence and diversity of trypanosomatids in bats from Mangabeiras Municipal Park (MMP), an urban park in Belo Horizonte, Minas Gerais, Brazil, providing insights into the role of urban environments in shaping the dynamics of these parasites.

Previous studies have reported the presence of *Leishmania* in sand flies in this park, suggesting a potential role of urban protected areas in maintaining trypanosomatid transmission cycles [19]. Belo Horizonte is endemic for CL and VL, with 416 human cases of VL and 663 cases of CL reported between 2020 and 2024 [20]. These findings highlight the need to investigate the role of bats in urban trypanosomatid transmission cycles and compare these dynamics with rural and sylvatic scenarios observed in other regions of Brazil and worldwide.

## Materials and methods

### Study area

Mangabeiras Municipal Park (MMP), located in the south-central region of Belo Horizonte, Minas Gerais, is considered one of the largest urban parks in Latin America, covering an area of 245.2 hectares. According to the Köppen-Geiger climate classification, the region has a subtropical highland climate (Cwb), characterized as humid temperate with dry winters [21]. MMP primarily features Atlantic Forest vegetation, with areas of Cerrado, especially on the

slopes, fostering rich biodiversity. Recent studies have assessed the presence of hemoparasites and ectoparasites in *Nasua nasua* (coatis) [22] and the detection of *Leishmania* in sand flies [19], suggesting the establishment of an epidemiological cycle in the area.

## Capture, processing, and collection of biological samples

Bats were captured using mist nets (12 m × 3 m) placed in an area accessible to park visitors. The sample collection site included a children's recreational area (Fig 1) surrounded by Semideciduous Seasonal Forest vegetation (19°57'03.3"S 43°54'30.4"W). Six sampling events were conducted in May, June (2), July (2), and December 2022, using 10 mist nets deployed from 6:00 PM to midnight, with inspections every 30 minutes.

After capture, the bats underwent a clinical examination, were classified according to age and sex, and identified using the classification keys of Vizotto and Taddei [23] and Gardner [24]. Captured animals classified as threatened according to the IUCN [25], as well as pregnant or lactating females, those with offspring, or those carrying identification bands, were immediately released.

Bats used in the study were anesthetized for blood collection via intracardiac puncture and subsequently euthanized for the collection of spleen, heart, liver, ear skin, and wing membrane samples. The collected material was stored in tubes containing 1X PBS supplemented with antibiotics (Penicillin 5,000 U/mL and Streptomycin 5,000 μg/mL) and antifungal agents (5-Fluorocytosine) for trypanosomatid isolation in culture medium, and in tubes containing RNAlater for molecular analysis.

## Culture medium

Biological samples of blood, tissues, and organs were macerated in sterile Petri plates using a scalpel blade and maintained at 25°C (±1°C) in biphasic NNN/LIT medium supplemented with 20% heat-inactivated sterile fetal bovine serum, antibiotics, and antifungal agents (as described above). Cultures were inspected weekly for two months for the observation of flagellated forms. Cultures showing fungal or bacterial contamination were discarded after macroscopic and/or microscopic evaluation using slide and coverslip examination.

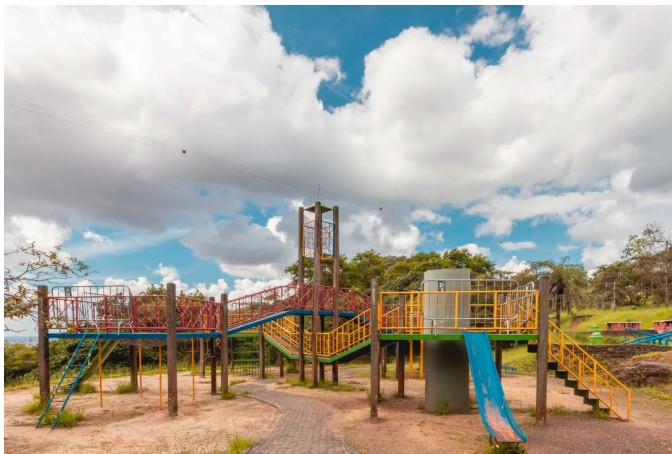

**Fig 1. Bat collection site within Mangabeiras Municipal Park (MMP), located in Belo Horizonte, Minas Gerais, Brazil.** Images under a CC BY license, with permission from Débora Cristina Capucci, copyright 2024.

### *Molecular detection of* **Trypanosomatidae**

DNA was extracted using the Gentra Puregene Tissue Kit® (Qiagen) for tissue samples, while the Blood GenomicPrep Mini Spin Kit (GE Healthcare) was used for blood samples. To assess the quality and integrity of the extracted genomic material, DNA samples were subjected to PCR targeting the cytochrome B (cytB) gene for blood samples, using the primers cytb1 (5'-CCATCCAACATCTCAGCATGATGAAA-3') and cytb2 (5'-GCCCCTCAGAATGA TATTTGTCCTCA-3') [26], and gamma-actin for tissues and organs, using the primers γ-fw (5'-ACAGAGAGAAGATGACGCAGATAATG-3') and γ-rv (5'-GCCTGAATGGCCACG TACA-3') [27]. For both reactions, 5 μL (10 ng/μL) of capybara (*Hydrochoerus hydrochaeris*) DNA, previously extracted by the research group, was used as a positive control.

The Nested-PCR technique, targeting the V7V8 region of the 18S gene (SSU rRNA), was used for the molecular detection of Trypanosomatidae. Reactions were performed using the primers TRY927F (5'-GAAACAAGAAACACGGGAG-3') and TRY927R (5'-CTACTGGG CAGCTTGGA-3'), amplifying 927 bp in the first reaction, and the primers SSU561F (5'-TGG GATAACAAAGGAGCA-3') and SSU561R (5'-CTGAGACTGTAACCTCAAAGC-3'), flanking 561 bp in the second reaction [28,29].

In all reactions, positive controls consisted of reference strains of *Leishmania amazonensis* (IFLA/BR/67/PH8), *Leishmania braziliensis* (MHOM/BR/75/M2903), *Leishmania infantum* (MHOM/BR/74/PP75), and *Leishmania guyanensis* (MHOM/BR/75/M4147). A negative control was also included, consisting of the PCR reagent mix without the addition of DNA template.

The amplified products were purified using the QIAquick PCR Purification Kit (Qiagen), and the amplicons were used for DNA sequencing under previously described conditions [30]. The obtained sequences were analyzed using FinchTV (Geospiza, Inc.) and MEGA X [31], and were compared with sequences deposited in GenBank using the BLAST tool. All sequences obtained were deposited in GenBank under the accession numbers OR656589 - OR656599.

### *Ethical statements*

This study was approved by the Animal Ethics Committee (CEUA) of the Oswaldo Cruz Foundation (protocol LW-3/23) and the Pontifical Catholic University of Minas Gerais (protocol 14/2022). Licenses for the collection of bats were granted by SISBIO (numbers 80543-4 and 82980-1). Specimens were deposited at the Natural Science Museum of the Pontifical Catholic University of Minas Gerais (MCN/CA-09/2022).

## Results

A total of 56 bats from the Phyllostomidae family were captured, representing seven species. The Stenodermatinae subfamily was the most represented, with 46 individuals (82.1%), followed by Carolliinae (6 individuals, 10.7%) and Glossophaginae (4 individuals, 7.1%). *Artibeus lituratus* was the most abundant species, with 28 individuals (50% of the total), followed by *Sturnira lilium* (N = 12, 21.4%) and *Carollia perspicillata* (N = 6, 10.7%) (Table 1).

A total of 149 samples, including blood (N = 32), tissues (ear skin and wing membrane, N = 18 each), and organs (liver, N = 40; heart, N = 41) from 52 individuals, were analyzed in NNN/LIT culture medium. Fungal or bacterial contamination was observed in 48 samples (32.2%), while the remaining 104 samples (67.8%) showed no trypanosomatid growth after two months of incubation.

The detection of trypanosomatids through the V7V8 Nested-PCR revealed 11 positive samples (5.9%) from nine animals out of a total of 186 samples (Fig 2A). The findings include: *Artibeus lituratus* presenting DNA of *Leishmania infantum* (N = 3) and *Trypanosoma* sp.

**Table 1. Species composition and distribution of bats captured by sampling month in Mangabeiras Municipal Park, 2022, Belo Horizonte, Minas Gerais, Brazil.**

| Species | Sampling months | | | | Total (%) |
|---|---|---|---|---|---|
| | May/2022 | Jun/2022 | Jul/2022 | Dec/2022 | |
| *Anoura caudifer* | 0 | 1 | 0 | 1 | 2 (3.6) |
| *Artibeus lituratus* | 9 | 2 | 10 | 7 | 28 (50.0) |
| *Artibeus obscurus* | 1 | 0 | 0 | 0 | 1 (1.8) |
| *Carollia perspicillata* | 1 | 2 | 3 | 0 | 6 (10.7) |
| *Glossophaga soricina* | 0 | 2 | 0 | 0 | 2 (3.6) |
| *Platyrrhinus lineatus* | 2 | 2 | 1 | 0 | 5 (8.9) |
| *Sturnira lilium* | 0 | 4 | 7 | 1 | 12 (21.4) |
| **Total (%)** | **13 (23.2)** | **13 (23.2)** | **21 (37.5)** | **9 (16.1)** | **56 (100)** |

The table lists the species of bats collected by month. The percentages represent the proportion of individuals among the total collected bats for each species and for each month.

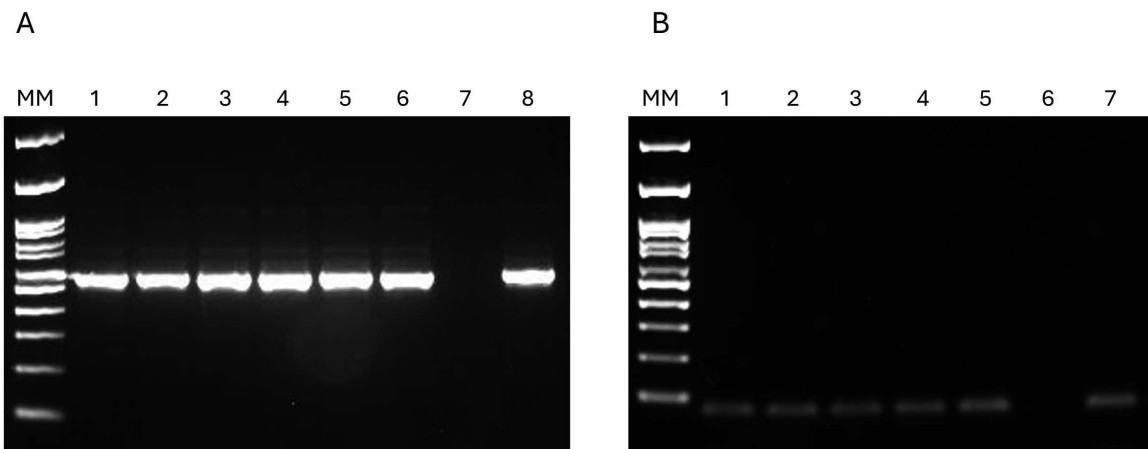

**Fig 2. Detection and validation of trypanosomatid DNA in bat samples.** (A) Representative agarose gel electrophoresis (2%) of V7V8 Nested PCR products showing samples positive for trypanosomatid DNA. MM: Molecular weight marker (100 bp ladder); Lanes 1-6: Positive bat samples; Lane 7: Negative control; Lane 8: Positive control (*L. braziliensis* reference strain). (B) Verification of genomic DNA integrity through the amplification of endogenous control genes. The PCR products of gamma-actin (70 bp) gene were used as endogenous controls to confirm the quality and integrity of the extracted genomic material. MM: Molecular weight marker (100 bp ladder); Lanes 1-5: Gamma-actin gene amplification products from tissue and organ samples; Lane 6: Negative control; Lane 7: Positive control (capybara DNA, *Hydrochoerus hydrochaeris*).

Neobat 3 (N = 1); *Anoura caudifer* carrying DNA of *Trypanosoma* sp. Neobat 4 (N = 1); *Carollia perspicillata* and *Glossophaga soricina* presenting DNA of *Leishmania infantum* (N = 1 each); *Sturnira lilium* presenting DNA of *Trypanosoma* sp. Neobat 3 (N = 1); and *Platyrrhinus lineatus* (N = 1) showing a mixed infection with DNA of *Leishmania infantum* and *Leishmania braziliensis* (Table 2).

Of the PCR-positive samples, seven (63.6%) yielded negative results in culture medium analysis, one (9.1%) showed contamination with fungi or bacteria, hindering evaluation for two months, and three (27.3%) were not collected. The endogenous amplification of *cytb* and *gamma-actin* genes demonstrated the integrity of the extracted genomic material (Fig 2B).

The phylogenetic analysis of the sequences confirmed the taxonomic positions of the isolates, placing them within the *Trypanosoma wauwau* clade (Fig 3A) and the *Leishmaniinae* subfamily (Fig 3B).

**Table 2. Detection of trypanosomatids in bat species from Mangabeiras Municipal Park, Belo Horizonte, Minas Gerais, Brazil, through V7V8 Nested-PCR.**

| Species | Trypanosomatidae/Sample* | | | | Total (%) |
|---|---|---|---|---|---|
| | *L. infantum* | *L. braziliensis* | *Trypanosoma* sp. Neobat 3 | *Trypanosoma* sp. Neobat 4 | |
| *Anoura caudifer* | 0 | 0 | 0 | 1[B] | **1(9,1)** |
| *Artibeus lituratus* | 2[H,L,S] | 0 | 1[B,L] | 0 | **3 (45,4)** |
| *Carollia perspicillata* | 1[L] | 0 | 0 | 0 | **1 (9,1)** |
| *Glossophaga soricina* | 1[S] | 0 | 0 | 0 | **1(9,1)** |
| *Platyrrhinus lineatus* | 1[H¶] | 1[B¶] | 0 | 0 | **2 (18,2)** |
| *Sturnira lilium* | 0 | 0 | 1[L] | 0 | **1(9,1)** |
| **Total (%)** | **5 (50.0)** | **1 (12.5)** | **2 (25.0)** | **1 (12.5)** | **9 (100)** |

The table lists the species of bats and the trypanosomatid infections identified. The symbols indicate the samples in which the parasites were detected: B = blood, H = heart, L = liver, S = spleen. Mixed infections are marked with a dagger (¶). The percentages represent the proportion of positive detections among the total infected samples for each species.

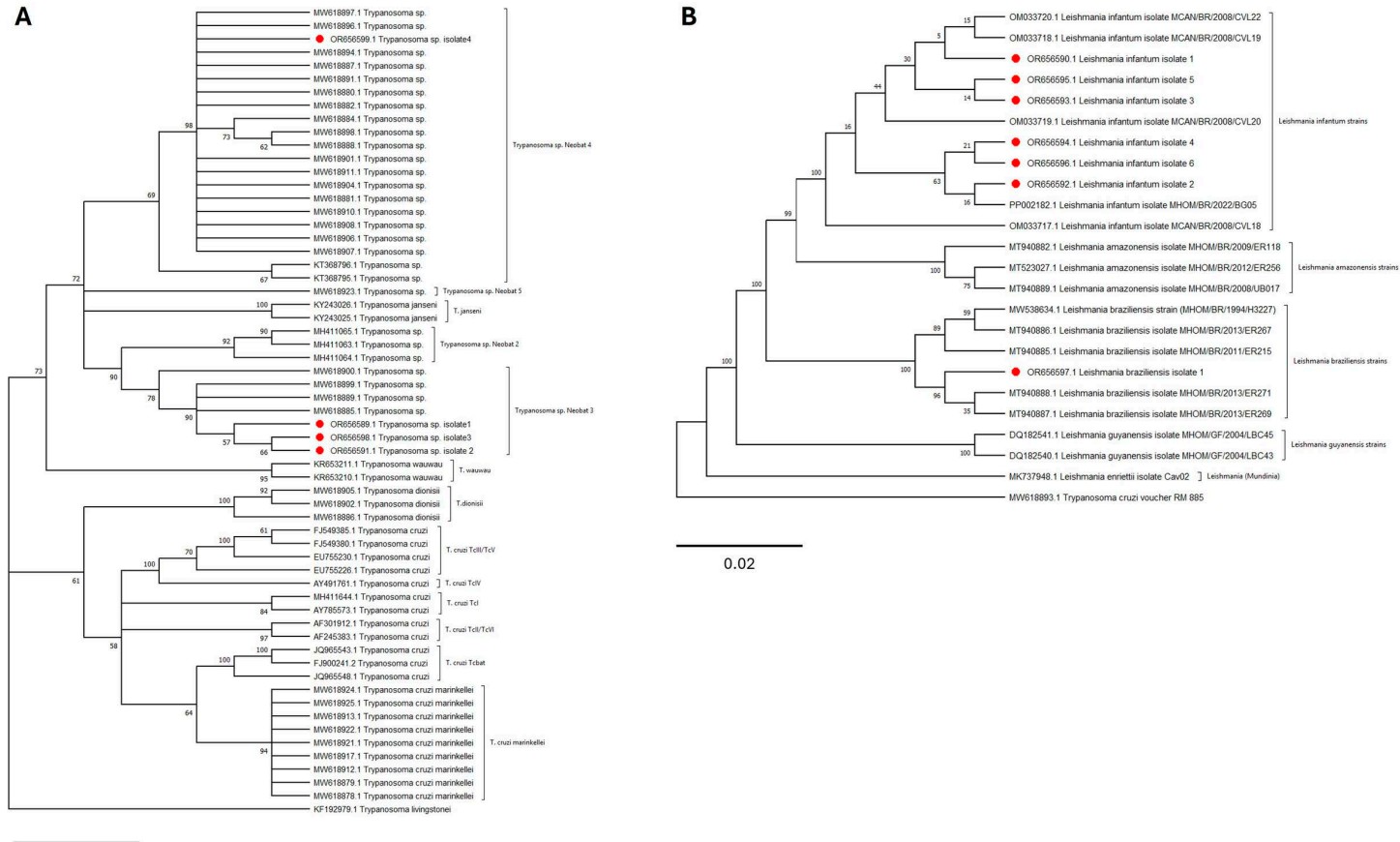

**Fig 3. Maximum likelihood phylogenetic trees of *Trypanosoma* and *Leishmania* isolates obtained from bats sampled in Mangabeiras Municipal Park, Brazil, based on 18S SSU rDNA sequences.** (A) Phylogenetic tree of *Trypanosoma* sp. Neobat sequences generated in this study, constructed using the Kimura 2-parameter (K2) model with 1,000 bootstrap replicates. (B) Phylogenetic tree showing the taxonomic positions of *Leishmania infantum* and *Leishmania braziliensis* sequences obtained in this study, constructed using the Kimura 2-parameter (K2) model with 1,000 bootstrap replicates. Bootstrap values (≥50%) are displayed at the nodes. Isolates from this study are highlighted in red circles, and reference sequences were retrieved from GenBank. Scale bars represent the number of substitutions per site.

## Discussion

In this study, the diversity of bat trypanosomes in MMP, one of the largest urban parks in Latin America, located in Belo Horizonte, Minas Gerais, Brazil, was highlighted through standard capture efforts. Molecular data revealed a predominance of *Leishmania infantum* DNA (50%), the etiological agent of VL. Additionally, sequences from two divergent kinetoplastid taxa were identified: one similar to *Trypanosoma* sp. Neobat 3 (25%) and another to *Trypanosoma* sp. Neobat 4 (12.5%), both considered hidden bat parasites. *Leishmania braziliensis* sequences were also detected in bats (12.5%). This study expands knowledge regarding the occurrence of trypanosomatids circulating in urban bats.

*Leishmania infantum* was previously detected in *Evandromyia edwardsi* from MMP [19], and the proximity of this urban park to residential neighborhoods where dogs are present underscores an epidemiological scenario involving this parasite in the urban area of Belo Horizonte. In this study, four bat species (*Artibeus lituratus*, *Carollia perspicillata*, *Glossophaga soricina*, and *Platyrrhinus lineatus*) were found carrying *L. infantum* DNA in different biological samples, such as heart, liver, and spleen. All these bats have been previously associated with *L. infantum* in Brazil [32], reinforcing their role in the transmission cycle.

The MOTUs designated as *Trypanosoma* spp. Neobat comprise a complex of unnamed trypanosomes divided into *T.* sp. Neobat 1-6 [33–36]. Each MOTU likely represents a valid species requiring formal taxonomic description. However, as these flagellates appear to be incapable of growing in conventional culture media for trypanosomatids [37], taxonomic classification remains incomplete. Therefore, combined strategies such as parasitological and molecular assays are essential for determining bat positivity.

To date, *T.* spp. Neobat gene sequences have been detected in phyllostomid bats from Panama, Colombia, and Brazil. The finding of *T.* sp. Neobat 3 in samples from *A. lituratus* in Belo Horizonte confirms previous reports from the state of Rio de Janeiro, where the parasite was also detected in this bat [35], reinforcing the putative role of the genus *Artibeus* as a reservoir of this trypanosomatid [33–35]. Additionally, *T.* sp. Neobat 3 was identified for the first time in *S. lilium*, a bat known to be a putative reservoir of *T. dionisii* [35,38], expanding knowledge on putative hosts.

*Trypanosoma* sp. Neobat 4, a member of the *T. wauwau* complex, was long considered a specialized parasite of the *Carollia* genus, particularly *C. perspicillata*, where the parasite was originally described [35]. However, *T.* sp. Neobat 4 has been identified in bats belonging to other genera, such as *Anoura caudifer*, *Glossophaga soricina*, and *Platyrrhinus recifinus*, disproving the previous hypothesis of host specificity to *Carollia* [38]. In the present study, sequences 100% similar to *T.* sp. Neobat 4 were confirmed in a blood sample from *A. caudifer*, indicating that other bats may, in fact, play a role in the transmission cycle of this parasite.

*Leishmania braziliensis* was previously found in *Evandromyia edwardsi* collected in caves from MMP, highlighting a putative transmission cycle involving this sand fly and other mammals [19]. In this study, *P. lineatus* was found to be positive for *L. braziliensis* in a blood sample, reinforcing previous findings [39]. Several Brazilian bats have been reported carrying *L. braziliensis*, especially insectivorous species such as *Cynomops planirostris* [40], *Eumops glaucinus* [41], *Molossus molossus* [42], *M. rufus* [41], and *Lasiurus cinereus* [40]. Although sand fly ingestion by bats is uncertain, and the probability of accidental transmission by grooming cannot be ruled out, the presence of *Leishmania* in frugivorous species [39,41], nectarivorous species [42,43], and hematophagous species [43] suggests alternative transmission routes, in addition to the traditional sand fly bite.

Knowledge gaps remain regarding the biological and ecological properties of MOTUs *T.* spp. Neobats, such as their vectors, transmission cycles, and interactions with hosts. Although sand flies are not considered natural vectors of *Trypanosoma*, naturally infected females with parasites

of this genus have been reported in Brazil. Among several sand fly species recorded in the study area [19], *Nyssomyia whitmani* has been associated with *Trypanosoma* sp. infection in Bahia [44]. Furthermore, representative species of the Shannoni series of the *Psathyromyia* genus have been found naturally infected with *Trypanosoma rangeli* and other unidentified trypanosomatids in Rondônia and Acre states, respectively [45]. However, a previous study on the sand fly fauna in the study area did not detect *Trypanosoma* infections in these species. While sand flies may play a role in the transmission of certain *Trypanosoma* species, more data are needed to establish their role in the transmission of *T.* spp. Neobats in urban environments.

In addition to sand flies, triatomine bugs, which are vectors of *Trypanosoma cruzi*, may also be involved in the transmission cycles of other trypanosomatids in bats. Indeed, the only known vector of *T. cruzi marinkellei* consists of triatomines from the genus *Cavernicola*, which are associated with bat colonies in caves, hollow trees, and palm trees [46,47]. Moreover, although the vector of *T. cruzi* Tcbat remains unknown, this genotype is most closely related to *T. cruzi* TcI, a group that is preferentially associated with didelphids and *Rhodnius* spp. in arboreal niches [48]. Similarly, the *T. rangeli* lineage TrE has only been found in bats and in the triatomine *Rhodnius pictipes* from Central and Amazonian regions of Brazil [49].

The mechanisms by which bats acquire these parasites remain poorly understood. Bats may acquire *T. cruzi* either through blood meals from infected triatomines or by ingesting these arthropods. Triatomines and bats may interact during nocturnal foraging, when both are active, or during the day, when bats roost in trees or caves, potentially allowing triatomines to feed on them [50]. This underscores the potential involvement of multiple vectors, such as sand flies and triatomines, in the zoonotic transmission cycles of *Trypanosoma* species, particularly in urban areas where bats, sand flies, and triatomines may ecologically overlap. Understanding these interactions is essential for elucidating the dynamics of zoonotic transmission and the role of bats as reservoirs for trypanosomatid parasites.

Interestingly, a mixed infection was detected in *P. lineatus*, with *L. infantum* detected in the heart and *L. braziliensis* in the blood. Although *L. braziliensis* seems to exhibit tropism for dermal tissues, the visceralization of *Viannia* species is considered common and transient, at least in animal models [51–55]. The occurrence of mixed natural infections with two or more *Leishmania* species is apparently rare but has been reported in bats [56] and rodents [57]. Moreover, mixed infections with *Trypanosoma* spp. have also been described [58,59], highlighting that the gregarious habits of bats may facilitate the transmission of these parasites.

Despite the limitations of this study, such as sampling in a single location within MMP and the parasitological monitoring of tissue and organ samples presenting a high level of fungal and bacterial contamination, this study demonstrates the occurrence of Trypanosomatidae species carried by bats in urban areas of Belo Horizonte. The cycles of *L. braziliensis* and *L. infantum* appear to be established in MMP, with the confirmation of sand flies and bats carrying these parasites. Further studies are needed to ascertain the role of *A. caudifer*, *A. lituratus*, and *S. lilium* in the transmission cycle of *T.* spp. Neobat and the involvement of putative vectors, such as kissing bugs.

## Acknowledgments

To the Fiocruz Network of Technological Platforms at Instituto René Rachou - Fiocruz Minas for the DNA sequencing facility, and their assistance and DNA sequencing services.

## Author contributions

**Conceptualization:** Mariana Alves Lima, Felipe Dutra-Rêgo, José Dilermando Andrade Filho.

**Data curation:** Mariana Alves Lima, Marcela Elisa Vaz, Ana Cristina Vianna Mariano da Rocha Lima.

**Formal analysis:** Mariana Alves Lima, Marcela Elisa Vaz, Ana Cristina Vianna Mariano da Rocha Lima, Débora Cristina Capucci, Felipe Dutra-Rêgo.

**Investigation:** Mariana Alves Lima, Marcela Elisa Vaz, Jennifer Emanuelle Ferreira.

**Methodology:** Mariana Alves Lima, Marcela Elisa Vaz, Jennifer Emanuelle Ferreira, Ana Cristina Vianna Mariano da Rocha Lima, Débora Cristina Capucci, Sônia Aparecida Talamoni, Felipe Dutra-Rêgo.

**Project administration:** José Dilermando Andrade Filho.

**Resources:** Sônia Aparecida Talamoni, José Dilermando Andrade Filho.

**Supervision:** Felipe Dutra-Rêgo, José Dilermando Andrade Filho.

**Validation:** Felipe Dutra-Rêgo.

**Visualization:** Mariana Alves Lima, Marcela Elisa Vaz, Felipe Dutra-Rêgo, José Dilermando Andrade Filho.

**Writing – original draft:** Mariana Alves Lima, Marcela Elisa Vaz, Felipe Dutra-Rêgo, José Dilermando Andrade Filho.

**Writing – review & editing:** Mariana Alves Lima, Marcela Elisa Vaz, Jennifer Emanuelle Ferreira, Ana Cristina Vianna Mariano da Rocha Lima, Débora Cristina Capucci, Sônia Aparecida Talamoni, Felipe Dutra-Rêgo, José Dilermando Andrade Filho.

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
