## [Decision Letter · Decision Letter 0]

22 Dec 2024

PONE-D-24-46291Hidden diversity of Trypanosomatidae (Protozoa: Kinetoplastea) in bats from an urban park in BrazilPLOS ONE

Dear Dr. Dutra-Rêgo,

Thank you for submitting your manuscript to PLOS ONE. After careful consideration, we feel that it has merit but does not fully meet PLOS ONE’s publication criteria as it currently stands. Therefore, we invite you to submit a revised version of the manuscript that addresses the points raised during the review process.

We look forward to receiving your revised manuscript.

Kind regards,

Vinícius Silva Belo

Academic Editor

PLOS ONE

2. We note that your Data Availability Statement is currently as follows: [All relevant data are within the manuscript and its Supporting Information files.] Please confirm at this time whether or not your submission contains all raw data required to replicate the results of your study. Authors must share the “minimal data set” for their submission. PLOS defines the minimal data set to consist of the data required to replicate all study findings reported in the article, as well as related metadata and methods (https://journals.plos.org/plosone/s/data-availability#loc-minimal-data-set-definition).

Please upload the completed Content Permission Form or other proof of granted permissions as an ""Other"" file with your submission

Additional Editor Comments:

Kindly address all points raised by the reviewers. Regarding the comments from Reviewer #1, please highlight the aspects that differentiate your study from previous research on the topic. Additionally, consider including information and analyses related to the ecological profile, as suggested.

Reviewers' comments:

Reviewer's Responses to Questions

**Comments to the Author**

1. Is the manuscript technically sound, and do the data support the conclusions?

Reviewer #1: Partly

Reviewer #2: Partly

Reviewer #3: Yes

2. Has the statistical analysis been performed appropriately and rigorously? 

Reviewer #1: N/A

Reviewer #2: N/A

Reviewer #3: Yes

3. Have the authors made all data underlying the findings in their manuscript fully available?

Reviewer #1: Yes

Reviewer #2: Yes

Reviewer #3: Yes

4. Is the manuscript presented in an intelligible fashion and written in standard English?

Reviewer #1: Yes

Reviewer #2: No

Reviewer #3: Yes

5. Review Comments to the Author

Reviewer #1: The work "Hidden diversity of Trypanosomatidae (Protozoa: Kinetoplastea) in bats from an urban park in Brazil" provides information about the occurrence of trypanosomatids in bats from an urban park in Belo Horizonte, Minas Gerais, Brazil. While the data is interesting as a record of zoonotic parasites in the study site, the scope of the information generated is limited to a national or regional level. The sample size is very small, and only a few bat species were examined. In Brazil, there are various studies on trypanosomatids in bats. In fact, infection with Leishmania in the bat species from this study has already been reported in Brazil and other South American countries (see Leishmania species infection of bats: A systematic review. Acta Tropica). No additional ecological or epidemiological information about these parasites in bats is provided. I suggest submitting the work to a national or regional journal.

Reviewer #2: The study under review clearly presents the main research question, which is to identify the presence of trypanosomatids in bats captured in Mangabeiras Municipal Park, Belo Horizonte, Brazil. The main findings, including the detection of different species of Leishmania and Trypanosoma, as well as the evaluation of the integrity of the extracted DNA, are well defined. The manuscript is scientifically solid, addresses a relevant topic for public health and ecology, and contributes to the understanding of the role of urban bats as potential reservoirs of parasites that cause neglected diseases, such as leishmaniasis and Chagas disease. However, there are aspects that could be improved to maximize the scientific impact and clarity of the work.

Points for Improvement

1. Abstract and Introduction

Contextualization and justification: Although the authors highlighted the relevance of ecological studies involving bats, it is not clear how this study advances or differs from previous works. It is recommended to:

- Include a more explicit analysis of the knowledge gap this work addresses.

- Add information about outbreaks or human cases in Belo Horizonte, connecting the findings to local epidemiology and comparing them with studies from other regions of Brazil or the world.

References: While relevant, it would be interesting to establish a more direct connection between recent studies (last five years) and the specific focus of the research to strengthen the justification.

Keywords: Including terms such as "epidemiology" or "zoonotic reservoirs" could improve the indexing and reach of the article.

2. Connection Between Methodology and Objective

Methodological explanation: The introduction addresses life cycles and transmission methods of the parasites but does not link these elements to the use of Nested-PCR and other techniques employed in the study. It is recommended to explain why the chosen methodologies are suitable for detecting trypanosomatids in bats.

3. Methodological Challenges and Data Presentation

Sample contamination: The contamination rate (32.2%) is acknowledged as a limitation, but it would be useful to explore more thoroughly how this contamination impacted the results.

Data inconsistencies:

- In the abstract, contamination is described as 31.5%, while in the results section, it is mentioned as 32.2%. This discrepancy should be corrected.

- In Table 2, the detection of trypanosomatids is reported in eight animals, but the text mentions nine. This error requires careful revision.

4. Discussion on Vectors

Detailing vectors: The relationship among bats, sand flies, and kissing bugs lacks sufficient detail, especially regarding potential transmission routes. It is recommended to expand this section, considering its relevance to zoonotic cycles.

5. Writing Revision

Language correction: Review the manuscript's writing in English, especially in Figure 1 and lines 8-11, which contain text written in Portuguese. It is recommended to adapt these sections to English to ensure consistency and standardization throughout the document.

Reviewer #3: Dear Authors,

Thank you for the opportunity to review the manuscript titled "Hidden diversity of Trypanosomatidae (Protozoa: Kinetoplastea) in bats from an urban park in Brazil." The study addresses a relevant and timely topic in parasitology, providing valuable data on the diversity of trypanosomatids in bats and their potential ecological and epidemiological implications. However, I identified several areas where improvements could be made to enhance the clarity and impact of the manuscript.

Below are some minor revisions:

Results

Line 169: “The detection of trypanosomatids through the V7V8 Nested-PCR revealed nine positive animals.” Change to "eight positive animals" or "nine positive samples."

Line 170: Fig. 2A: This does not match the nine positive animals; however, changing to "nine positive samples from six animals" would be correct.

The sequencing analysis is absent; including a dendrogram or phylogenetic tree to locate the sequences of the trypanosomatids found would strengthen the manuscript. Note that sequencing results are not discussed in the text.

I believe these suggestions will contribute to improving the manuscript, making it more robust and clear. I congratulate the authors on their work.

Sincerely.

6. PLOS authors have the option to publish the peer review history of their article (what does this mean? ). If published, this will include your full peer review and any attached files.

**Do you want your identity to be public for this peer review?** For information about this choice, including consent withdrawal, please see our Privacy Policy .

Reviewer #1: No

Reviewer #2: No

Reviewer #3: No

---

## [Author Response · Author response to Decision Letter 0]

20 Jan 2025

Review Comments to the Author

Reviewer #1:

General comment:

The work "Hidden diversity of Trypanosomatidae (Protozoa: Kinetoplastea) in bats from an urban park in Brazil" provides information about the occurrence of trypanosomatids in bats from an urban park in Belo Horizonte, Minas Gerais, Brazil. While the data is interesting as a record of zoonotic parasites in the study site, the scope of the information generated is limited to a national or regional level. The sample size is very small, and only a few bat species were examined. In Brazil, there are various studies on trypanosomatids in bats. In fact, infection with Leishmania in the bat species from this study has already been reported in Brazil and other South American countries (see Leishmania species infection of bats: A systematic review. Acta Tropica). No additional ecological or epidemiological information about these parasites in bats is provided. I suggest submitting the work to a national or regional journal.

Authors’ comment: Thank you for your comments. We understand that three major points were raised, and we would like to address each of them below:

1- Sample size:

While we acknowledge that the sample size could be larger, we respectfully disagree with the characterization of it as “very small”. In prospective studies like ours, analyzing the highest number of samples possible is ideal to ensure representativeness. However, in urban areas such as the study site, the bat population density tends to be lower compared to wild and rural areas, which naturally impacts sample availability. Additionally, logistical challenges, including restrictions imposed by the COVID-19 pandemic during the study period, limited the feasibility of additional collections.

It is important to note that sample sizes in studies on trypanosomatids in bats vary widely in the literature, with some studies reporting even smaller sample sizes than ours. We believe the data presented here provides a meaningful snapshot of the local ecological scenario, acknowledging that additional species of bats and trypanosomatids might not have been captured. This limitation, while recognized, does not diminish the value of the findings.

2- Presence of Leishmania infections already reported in Brazil and absence of ecological/epidemiological information:

As shown in Table 2, our study does not focus solely on Leishmania infections, which were reported in 6 out of 9 bats. We also documented infections by Trypanosoma sp. Neobat in 3 bats. This finding is noteworthy, as there is limited information available on T. sp. Neobat, particularly in urban areas. The presence of this parasite in a protected urban area of Belo Horizonte represents a novel contribution to the field.

Additionally, this is the first well-documented study on bat ecology in the protected urban areas of Belo Horizonte investigating trypanosomatid infections, further reinforcing its novelty. Regarding the ecological and epidemiological data, we have expanded this section in the revised manuscript to include more context on these aspects (lines 71-88). However, as acknowledged, knowledge gaps regarding T. sp. Neobat remain, as highlighted by several authors (e.g., Alves et al., 2021; DOI: https://doi.org/10.1016/j.ijppaw.2021.09.003).

3- Regional impact of the results:

While the findings have a regional focus, we believe they provide valuable contributions to the broader understanding of trypanosomatids in bats. For instance, we expanded the geographical distribution and ecological knowledge of T. sp. Neobat, recording it for the first time in the state of Minas Gerais. This adds to the limited literature on this parasite, particularly in urban settings, and is of relevance to high-impact journals such as PLOS ONE.

Reviewer #2:

General comment:

The study under review clearly presents the main research question, which is to identify the presence of trypanosomatids in bats captured in Mangabeiras Municipal Park, Belo Horizonte, Brazil. The main findings, including the detection of different species of Leishmania and Trypanosoma, as well as the evaluation of the integrity of the extracted DNA, are well defined. The manuscript is scientifically solid, addresses a relevant topic for public health and ecology, and contributes to the understanding of the role of urban bats as potential reservoirs of parasites that cause neglected diseases, such as leishmaniasis and Chagas disease. However, there are aspects that could be improved to maximize the scientific impact and clarity of the work.

Specific comments:

Abstract and Introduction

Contextualization and justification: Although the authors highlighted the relevance of ecological studies involving bats, it is not clear how this study advances or differs from previous works. It is recommended to:

- Include a more explicit analysis of the knowledge gap this work addresses.

Authors’ comment: We included your suggestion in this new version of the manuscript (lines 71-88).

- Add information about outbreaks or human cases in Belo Horizonte, connecting the findings to local epidemiology and comparing them with studies from other regions of Brazil or the world.

Authors’ comment: We included your suggestion in this new version of the manuscript (lines 83-85).

References: While relevant, it would be interesting to establish a more direct connection between recent studies (last five years) and the specific focus of the research to strengthen the justification.

Authors’ comment: We have included your suggestion in this new version of the manuscript.

Keywords: Including terms such as "epidemiology" or "zoonotic reservoirs" could improve the indexing and reach of the article.

Authors’ comment: We have included "Epidemiology" and "Disease Reservoirs" as keywords. The term "zoonotic reservoir" is not a MeSH term, and therefore, we have avoided using it as a keyword.

Connection Between Methodology and Objective

Methodological explanation: The introduction addresses life cycles and transmission methods of the parasites but does not link these elements to the use of Nested-PCR and other techniques employed in the study. It is recommended to explain why the chosen methodologies are suitable for detecting trypanosomatids in bats.

Authors’ comment: We have included your suggestion in this revised version of the manuscript (lines 63-66).

Methodological Challenges and Data Presentation Sample contamination: The contamination rate (32.2%) is acknowledged as a limitation, but it would be useful to explore more thoroughly how this contamination impacted the results.

Authors’ comment: We have incorporated your suggestion into the results section (lines 194-197) and the final paragraph of the discussion, where the study's limitations are addressed (lines 286-293).

Data inconsistencies:

- In the abstract, contamination is described as 31.5%, while in the results section, it is mentioned as 32.2%. This discrepancy should be corrected.

Authors’ comment: This discrepancy has now been corrected.

- In Table 2, the detection of trypanosomatids is reported in eight animals, but the text mentions nine. This error requires careful revision.

Authors’ comment: Table 2 has been updated and corrected in this revised version of the manuscript.

Discussion on Vectors

Detailing vectors: The relationship among bats, sand flies, and kissing bugs lacks sufficient detail, especially regarding potential transmission routes. It is recommended to expand this section, considering its relevance to zoonotic cycles.

Authors’ comment: In this revised version of the manuscript, we have expanded the discussion on the role of potential vectors in transmitting trypanosomatids (lines 256–277).

Writing Revision

Language correction: Review the manuscript's writing in English, especially in Figure 1 and lines 8-11, which contain text written in Portuguese. It is recommended to adapt these sections to English to ensure consistency and standardization throughout the document.

Authors’ comment: The manuscript has been thoroughly reviewed for English language accuracy.

Reviewer #3:

General comment:

Thank you for the opportunity to review the manuscript titled "Hidden diversity of Trypanosomatidae (Protozoa: Kinetoplastea) in bats from an urban park in Brazil." The study addresses a relevant and timely topic in parasitology, providing valuable data on the diversity of trypanosomatids in bats and their potential ecological and epidemiological implications. However, I identified several areas where improvements could be made to enhance the clarity and impact of the manuscript.

Specific comments:

Line 169: “The detection of trypanosomatids through the V7V8 Nested-PCR revealed nine positive animals.” Change to "eight positive animals" or "nine positive samples."

Authors’ comment: This paragraph has been changed (lines 187-188).

Line 170: Fig. 2A: This does not match the nine positive animals; however, changing to "nine positive samples from six animals" would be correct.

Authors’ comment: Figure 2, panels A and B, represent agarose gel electrophoresis of V7V8 and gamma-actin PCR products, respectively, with a subset of positive samples displayed.

The sequencing analysis is absent; including a dendrogram or phylogenetic tree to locate the sequences of the trypanosomatids found would strengthen the manuscript. Note that sequencing results are not discussed in the text.

Authors’ comment: Thank you for your comment. All PCR-positive samples were 100% identical to the corresponding sequences in GenBank. In this context, constructing a phylogenetic tree would not significantly add novelty to the manuscript, as the phylogenetic positions of all parasites identified in our study are already well-established. Typically, sequencing data in such cases are presented descriptively. However, we have included Figure 3 with the phylogenetic analysis for further clarity and completeness.

---

## [Decision Letter · Decision Letter 1]

6 Feb 2025

PONE-D-24-46291R1Hidden diversity of Trypanosomatidae (Protozoa: Kinetoplastea) in bats from an urban park in BrazilPLOS ONE

Dear Dr. Dutra-Rêgo,

Thank you for submitting your manuscript to PLOS ONE. After careful consideration, we feel that it has merit but does not fully meet PLOS ONE’s publication criteria as it currently stands. Therefore, we invite you to submit a revised version of the manuscript that addresses the points raised during the review process.

We look forward to receiving your revised manuscript.

Kind regards,

Vinícius Silva Belo

Academic Editor

PLOS ONE

Journal Requirements:

Reviewers' comments:

Reviewer's Responses to Questions

**Comments to the Author**

1. If the authors have adequately addressed your comments raised in a previous round of review and you feel that this manuscript is now acceptable for publication, you may indicate that here to bypass the “Comments to the Author” section, enter your conflict of interest statement in the “Confidential to Editor” section, and submit your "Accept" recommendation.

Reviewer #2: All comments have been addressed

Reviewer #3: All comments have been addressed

2. Is the manuscript technically sound, and do the data support the conclusions?

Reviewer #2: Yes

Reviewer #3: Yes

3. Has the statistical analysis been performed appropriately and rigorously? 

Reviewer #2: N/A

Reviewer #3: Yes

4. Have the authors made all data underlying the findings in their manuscript fully available?

Reviewer #2: Yes

Reviewer #3: Yes

5. Is the manuscript presented in an intelligible fashion and written in standard English?

Reviewer #2: No

Reviewer #3: Yes

6. Review Comments to the Author

Reviewer #2: The manuscript "Hidden diversity of Trypanosomatidae in bats from an urban park in Brazil" presents a solid study on trypanosomatid diversity in urban bats in Belo Horizonte. The authors addressed previous concerns, enhancing the manuscript’s clarity, methodology, and scientific rigor. The study contributes valuable insights into the role of bats as reservoirs for diseases like leishmaniasis and Chagas disease.

Specific comments:

Abstract & Introduction:

The knowledge gap is clearly defined (lines 71–88), with added local epidemiological data (lines 83–85) and relevant recent references. Keywords were updated appropriately.

Methodology:

The rationale for using Nested-PCR is well-explained (lines 63–66). Capture methods and ethical protocols are detailed, and the impact of the 32.2% contamination rate is thoughtfully discussed (lines 194–197, 286–293).

Results & Data Consistency:

Data inconsistencies have been corrected, including contamination rates and Table 2 updates. Tables and figures are clear and accurate.

Discussion on Vectors:

Expanded discussion on vector roles (lines 256–277), with a minor suggestion to add more references on triatomine-bat interactions.

Language & Style:

The language issues have been addressed, with minor suggestions to improve the text's fluency. Additionally, some sections are still written in Portuguese and should be translated to ensure consistency throughout the manuscript.

Recommendation:

Accept with Minor Revisions. The manuscript is well-prepared, scientifically robust, and contributes meaningful data. Minor language adjustments and additional references will further strengthen the paper.

Reviewer #3: (No Response)

7. PLOS authors have the option to publish the peer review history of their article (what does this mean? ). If published, this will include your full peer review and any attached files.

**Do you want your identity to be public for this peer review?** For information about this choice, including consent withdrawal, please see our Privacy Policy .

Reviewer #2: No

Reviewer #3: No

---

## [Author Response · Author response to Decision Letter 1]

6 Feb 2025

Reviewer #2:

Accept with Minor Revisions. The manuscript is well-prepared, scientifically robust, and contributes meaningful data. Minor language adjustments and additional references will further strengthen the paper.

Authors’ comments: We conducted a thorough review of the manuscript, addressing typographical errors and misspellings. Additionally, we have incorporated additional references related to triatomine-bat interactions, as requested.

---

## [Editor Report · Decision Letter 2]

9 Feb 2025

Hidden diversity of Trypanosomatidae (Protozoa: Kinetoplastea) in bats from an urban park in Brazil

PONE-D-24-46291R2

Dear Dr. Dutra-Rêgo,

We’re pleased to inform you that your manuscript has been judged scientifically suitable for publication and will be formally accepted for publication once it meets all outstanding technical requirements.

Kind regards,

Vinícius Silva Belo

Academic Editor

PLOS ONE

Additional Editor Comments (optional):

Congratulations to the authors on their insightful work. It is a pleasure to accept your manuscript for publication!
---

## [Editor Report · Acceptance letter]

PONE-D-24-46291R2

PLOS ONE

Dear Dr. Dutra-Rêgo,

I'm pleased to inform you that your manuscript has been deemed suitable for publication in PLOS ONE. Congratulations! Your manuscript is now being handed over to our production team.

Kind regards,

on behalf of

Dr. Vinícius Silva Belo

Academic Editor

PLOS ONE